# MazeNet:
# An Accurate, Fast, & Scalable Deep Learning Solution For Steiner Minimum Trees

## Abstract

The Obstacle Avoiding Rectilinear Steiner Minimum Tree (OARSMT) problem, which seeks the shortest interconnection of a given number of terminals in a rectilinear plane while avoiding obstacles, is a critical task in integrated circuit design, network optimization, and robot path planning. Since OARSMT is NP-hard, exact algorithms scale poorly with the number of terminals, leading practical solvers to sacrifice accuracy for large problems. However, for smaller-scale environments, there is no justification for failing to discover the true shortest path. To address this gap, we propose and study *MazeNet*, a deep learning-based method that learns to solve the OARSMT from data. MazeNet reframes OARSMT as a maze-solving task that can be addressed with a recurrent convolutional neural network (RCNN). A key hallmark of MazeNet is its ability to generalize: we only need to train the RCNN blocks on mazes with a small number of terminals; mazes with a larger number of terminals can be solved simply by replicating the same pre-trained blocks to create a larger network. Across a wide range of experiments, MazeNet achieves perfect OARSMT-solving accuracy with substantially reduced runtime compared to classical exact algorithms, and its perfect accuracy ensures shorter path lengths compared to state-of-the-art approximation algorithms.

## 1 Introduction

Maze-solving algorithms are a special case of the solutions of an Obstacle-Avoiding Rectilinear Steiner Minimum Tree (OARSMT), with important applications in fields such as integrated circuit design (Kahng et al., 2022), routing networks (Dong et al., 2013), and multi-path planning for robotics (Choset et al., 2005) and (Zang et al., 2022). In these domains, even small improvements in path lengths can unlock new operational capabilities and reduce costs. For instance, in Very Large Scale Integration (VLSI) design, minimizing the overall length of wiring directly decreases the power consumption, reduces signal congestion, and minimizes timing delays; all of which contribute to enhanced system performance Bricaud (2002). In this case, a maze can be expressed as a graph where all edges have uniform cost, and therefore OARMST directly translates to reducing the number of edges that are required to connect all the target nodes, which are termed terminals.

The Rectilinear Steiner Minimum Tree (RSMT) problem aims to connect a given set of points using only horizontal and vertical lines while minimizing the total connection length Hwang et al. (1992). OARSMT extends this problem by adding obstacles that must be avoided by the path solutions. RSMT is however NP-complete (Garey & Johnson, 1977), and the inclusion of obstacles further increases the problem's complexity. Therefore, an exhaustive graph approach (i.e., a search algorithm that mimics the exact solution of the Traveling Salesman Problem (Gutin & Punnen, 2002)) will be perfectly accurate, but will have poor runtime scaling with an increasing number of terminals. This method involves permuting the terminals and using Dijkstra's algorithm Cormen et al. (2009) to compute the shortest path between each pair of consecutive terminals. We will refer to this method as Dijkstra's exhaustive throughout our work.

Approximation algorithms to solve OARMST, such as by Kou et al. (Kou et al., 1981) and Mehlhorn (Mehlhorn, 1988), offer improved scaling. However, these methods are not guaranteed to be exact and can produce incorrect maze solutions. Figure 1 illustrates a maze where approximation methods

yield an incorrect result when trying to find the shortest path connecting three terminals, due to multiple paths having nearly equal lengths. We observe that even for this relatively small $11 \times 11$ node graph represented by a maze, accuracy is a serious issue for approximation algorithms.

Recent advancements in machine learning have introduced innovative approaches to tackling the Obstacle-Avoiding Rectilinear Minimum Steiner Tree (OARMST) problem. Among these,(Xiao et al., 2023), a mixed neural-algorithmic framework, offers a novel solution but does not account for obstacles in its computations. Similarly, while the methods proposed by (Chen et al., 2022), (Lin et al., 2023), (Huang & Young, 2011) ,and (Chu & Wong, 2008) address obstacle-avoiding Steiner tree construction, they are tailored for VLSI layouts and are not readily applicable to maze-like configurations. This constraints our comparison to explicitly graph approximation methods that handle more generalized environments, such as mazes with very specific obstacle patterns.

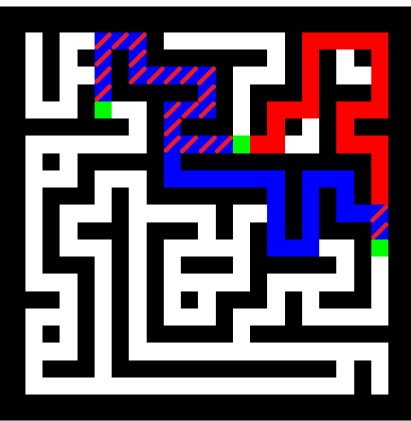

Figure 1: A maze with three terminals (green squares), with an incorrect approximation solution that is not the minimum-length path (red), and the correct exhaustive solution (blue). The overlap between the solutions is shown in blue with red stripes.

Rather than approaching OARMST as a traditional graph problem which has limitations of scalability, accuracy and generalization, we instead view mazes as images, and solve them via image processing techniques. Our deep neural network method, which we call MazeNet, makes use of Recurrent Convolutional Neural Networks (RCNNs), which have been previously used to solve mazes of only two terminals Schwarzschild et al. (2021), (Bansal et al., 2022). MazeNet is scalable and extendable, and can be used to solve mazes of variable sizes and variable numbers of terminals to be connected: in this paper, we present results for mazes of up to a size of $11 \times 11$ and up to eight terminals. MazeNet is able to offer perfect empirical accuracy for these scenarios, despite being trained on smaller mazes that have up to five terminals.

Since RCNNs may not be able to recognize a correct solution and terminate, MazeNet incorporates a search-based algorithm that recognizes a correct solution to the maze. Our method combines the runtime efficiency of graph-based approximate algorithms with the accuracy of the graph-based exhaustive algorithms. This work addresses the gap between probabilistically correct approximation methods and the demand for deterministic accuracy in small-scale environments. For problems within the 11×11 regime, probabilistic methods may be effective, but they lack justification for failing to discover the true shortest path. MazeNet is designed to achieve deterministic accuracy comparable to exact graph-based methods while maintaining the runtime efficiency of approximation algorithms.

The paper is organized as follows. In Section 2, we formally define the OARSMT problem and present its computational complexity. In Section 3, we present MazeNet's architecture and training process, highlighting its key algorithmic features. In Section 4, we present the performance of MazeNet and its state-of-the-art alternatives with regards to their accuracies and runtimes, and analyze their scalabilities. We conclude in Section 5 and discuss future work and extensions for MazeNet, situating its broader significance in regards to Deep Learning for complex problems.

## 2  PROBLEM STATEMENT

The OARSMT problem is formally defined as follows. Given a set of terminals $T = \{t_1, t_2, \ldots, t_N\}$ in a 2D plane, where each terminal $t_i$ has coordinates $(x_i, y_i)$, and a set of rectangular obstacles $O = \{O_1, O_2, \ldots, O_m\}$, with each obstacle $O_j$ defined by its vertices $(x_{j\,\min}, y_{j\,\min})$ and $(x_{j\,\max}, y_{j\,\max})$, the objective is to find a tree $T'$ that connects all terminals in $T$ using horizontal and vertical line segments (rectilinear paths) while avoiding all obstacles in $O$, such that the total length of the tree $T'$ is minimized. Defining $E(T')$ as the set of edges in the tree $T'$, and $L(T')$ as the total length of the tree, we have:

$$T' = \arg\min_{T'} L(T') = \sum_{e \in E(T')} \text{length}(e), \tag{1}$$

such that $T'$ is a connected acyclic graph spanning all terminals in $T$, and for every edge $e \in E(T')$ between two points $(x_i, y_i)$ and $(x_j, y_j)$, the path is rectilinear and not intersecting any $O_j \in O$.

With an exhaustive method, all possible permutations of the $T$ terminals are considered to evaluate every possible sequence of connections. For each permutation, the shortest path is computed between each sequential pair of terminals in the order specified by the permutation. The number of such permutations is $O(T!)$. Dijkstra's algorithm is used to compute the shortest path for each sequential pair, with a time complexity of $O(V \log V + E)$ (Cormen et al., 2009), where $V$ is the number of vertices and $E$ is the number of edges. In a grid graph with $V$ vertices and $E \approx V$ edges, this simplifies to $O(V \log V)$. Therefore, the total complexity of Dijkstra's exhaustive method is $O(T! \times (V + 1) \log V)$.

### 2.1  TRANSFORMING GRAPHS INTO IMAGES

To transform this graph-based problem into an image domain, we first note that mazes can be constructed using the Depth-First Search (DFS) algorithm Sibeyn et al. (2001), with each node in the maze connected to a set of neighbors in a lattice structure. Walls in the maze correspond to obstacles in the OARSMT problem. We add cycles to this DFS-generated maze by randomly removing some of the walls. We then place the $N$ terminals to be connected at random, uniformly selected locations across the maze. Detailed maze generation steps are provided in Appendix A.1.

As an example of this procedure's usage in MazeNet, consider the $11 \times 11$ node grid with 10 edges connecting nodes in Figure 2a, resulting in a total of $21 \times 21$ "cells". In the image representation of this graph in Figure 2b, each cell is a $2 \times 2$ pixel square with a 3-pixel padding around the maze, with the walls and the padding in black and other pixels in white. Thus, Figure 2b is a $48 \times 48$ pixel image representing the 121-node graph of Figure 2a. For the eventual purpose of training a neural network, when we extract the shortest path through the maze in Figure 2b, we generate the training output image in Figure 2c.

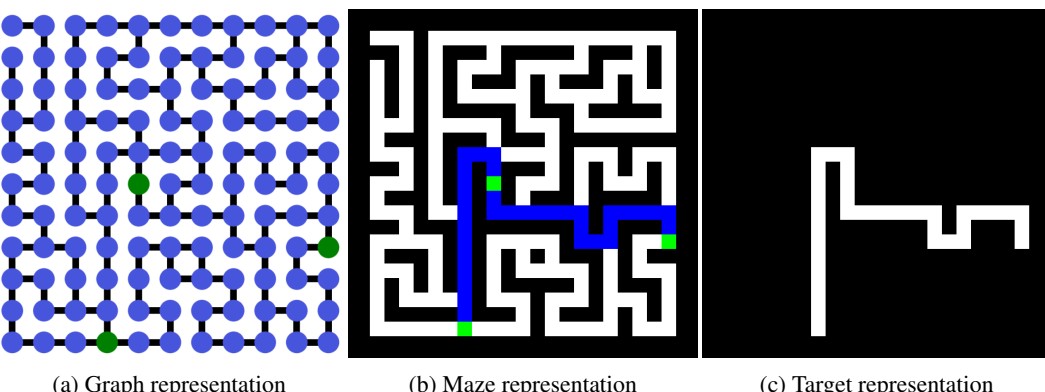

(a) Graph representation   (b) Maze representation   (c) Target representation

Figure 2: Example of converting a graph (a) to its corresponding maze (b), with the target image (c) as its shortest-path solution.

# 3 OVERVIEW OF MAZENET

We now present the proposed MazeNet method, starting with its Recurrent Convolutional Neural Network (RCNN) building blocks in Section 3.1. In Section 3.2, we introduce the termination condition module and present its algorithm in detail. In Section 3.3, we present the overall MazeNet architecture and its training procedure. Finally, in Section 3.4, we describe an algorithmic parallelization feature that is useful for MazeNet's extension to very large mazes.

## 3.1 RCNN APPROACH

The building block of the MazeNet method is the RCNN, which has been widely applied to object recognition tasks Liang & Hu (2015). RCNNs can processes images by applying convolution operations recurrently, focusing on learning scalable algorithms without any graph as an input. Separately, maze-inspired algorithms have long been employed to tackle the OARSMT problem Lin et al. (2018); yet there has been little connection between RCNN-based methods and graph-theoretic approaches. Schwarzschild et al. (2021) and Bansal et al. (2022) were the first to bridge this gap by using RCNNs to solve maze-related problems, mimicking traditional algorithmic processes. However, these problems were in domains where traditional methods are both fast and accurate, leaving open the question of whether RCNNs can provide similar advantages for more complex graph-based problems.

Since RCNNs are used for supervised learning in our case, it is necessary to have pairs of training input and target data images. We start by creating graphs as described in Appendix A.1, using the DFS algorithm followed by random wall removal and uniformly distributing N terminals across the grid. We then generate the solution graph with Dijsktra's exhaustive, a graph-based algorithm as described in Section 2, followed by a transformation to images as detailed in Section 2.1 in order to create the input/target image pairs in Figures 2b and 2c.

The architecture and the progressive training algorithm of Bansal et al. (2022) were designed to connect two terminals in perfect mazes, i.e., where a unique solution connects any two nodes. This architecture has three key stages: a projection module that processes the input, followed by a recurrent module (RB) that operates sequentially, and finally, a head module that produces the output (Appendix A.2). At each step of the recurrent module, a skip connection is maintained from the input, ensuring that the input information is preserved throughout the recurrence. A final head module transforms the network's output into a single-channel prediction. The width of the network, defined by the number of channels, is a tunable hyperparameter which represents the number of filters of the recurrent module. In this work, we use their progressive training algorithm and make modifications to the architecture to adapt it to this new problem, as detailed in Sections 3.3 and 3.2

RCNNs offer step-by-step interpretability of a trained method's operations, since the head module can be applied after any iteration, making it possible to observe the intermediate stages of the solution. These stages can be visualized directly as image outputs for insights into the solution process, as demonstrated in Figure 3. Here, after suppressing the white permissible paths of the maze in the input image, exploratory paths originate from the terminals, which intensify and become stable when connections are established. Note however that the maze is essentially solved by iteration 23 from a simple visual examination, and one does not have to wait for the paths to become solidly white. This observation motivates the introduction of a termination condition which can automate the recognition of the maze being essentially solved, and to return this correct solution, as discussed next.

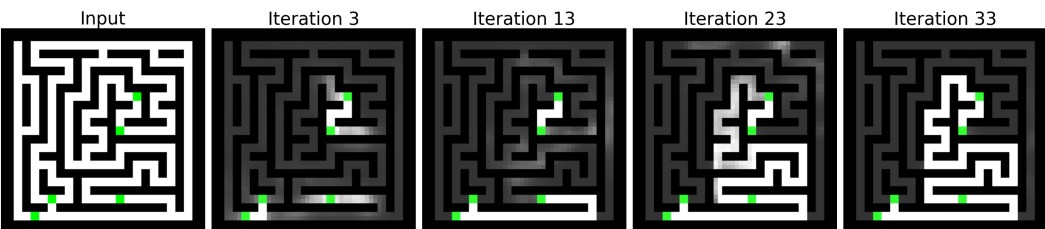

Figure 3: Visualization of MazeNet's progress over time as it connects four terminals.

## 3.2 TERMINATION CONDITION

The RCNN architecture of Bansal et al. (2022) lacks a termination condition within its recurrent modules, so its number of iterations must be predefined; which can lead to premature termination or to excessive runtimes. Another issue is that if there are two nearly equidistant paths, the output frequently oscillates between them. This event is also problematic with the approximation algorithms Kou (Kou et al., 1981) and Mehlhorn (Mehlhorn, 1988). However, during this oscillation the correct solution is occasionally highlighted as in Figure: 4. We therefore introduce a Termination Condition (TC) module, which reduces unnecessary iterations by halting the network's operation once the correct path is identified, and also achieves the maximum empirical accuracy. TC does incur a computational overhead, making its optimization crucial.

After a set number of iterations, the head module is followed by the TC module which performs a guided search, assessing the "whiteness" of the intermediate output which is measure of the presence of white pixels that represent the connected path. Starting at the upper leftmost terminal in the maze, TC specifically evaluates the "whiteness" in a $2 \times 2$ neighboring cell grid, averaging the 0-1 value of the 4 pixels at neighboring cell. If the whiteness exceeds an empirically determined threshold of 0.65, TC will continue the exploration in any direction North, East, West, and South (NEWS) above the threshold.

When encountering a junction where multiple directions exceed the whiteness threshold, TC initiates a recursive branch of exploration while keeping track of the main branch with a from-junction variable. If a branch leads to a previously visited position, indicating a cycle, the TC module stops, and the image undergoes additional iterations to ensure valid tree convergence. A last-move variable keeps track of the previous direction to avoid moving backwards during the exploration.TC maintains a record of visited positions, to ensure that all the terminals are reached or to terminate if no valid exploration direction remains.

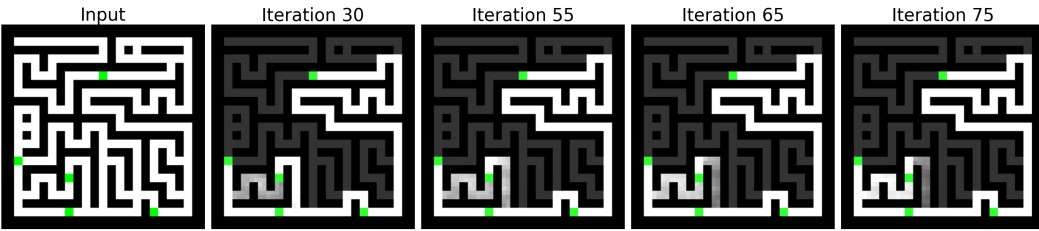

Figure 4: Visualization of progress when a termination condition is needed to resolve between two almost equidistant paths, since otherwise MazeNet oscillates between them.

---

**Algorithm 1** Implementation Details of the Termination Condition (TC) Module

---

**Input:** Initial position, image tensor, directions
from_junction = False
last_move = None
**while** all green terminals not visited
  **if** position revisited **then**
    **return** False
  Calculate "whiteness" for each direction (NEWS) around the current position
  **if** "whiteness" $> 0.65$ and opposite last_move is not the direction **then**
    Move to the direction with the highest "whiteness"
    last_move = highest whiteness direction
  **if** junction found (two or more available directions excluding last_move) **then**
    **if** from_junction is True and junction already explored **then**
      **return** False
    from_junction = True
    Recursively explore each direction from the junction
  **if** no valid move **then**
    **return** False
**return** True

---

Algorithm 1 enables the TC module to identify the correct path, halting MazeNet once it confirms that the maze is solved. Although this adds a runtime overhead of up to 40% on average, we have found it necessary for achieving an empirical 100% accuracy on our datasets with MazeNet. Since TC is computationally expensive, we avoid applying it after every iteration of the RB module to minimize overhead. Thus, we can conceptually condense the recurrent iterations that take place before each application of TC into a Batch module. Empirically, we set the first Batch to represent a total of 30 RB module iterations, and each subsequent Batch to represent 10 iterations.

### 3.3 COMPLETE MAZENET ARCHITECTURE

MazeNet combines the head module in Section 3.1 and the TC and Batch modules in Section 3.2. This architecture, depicted in Figure 5, has each of its modules composed of traditional 2D image operations such as convolutions, ReLU activations, skip connections, and Argmax operations. Refer to Appendix A.2 for a more detailed overview of each module.

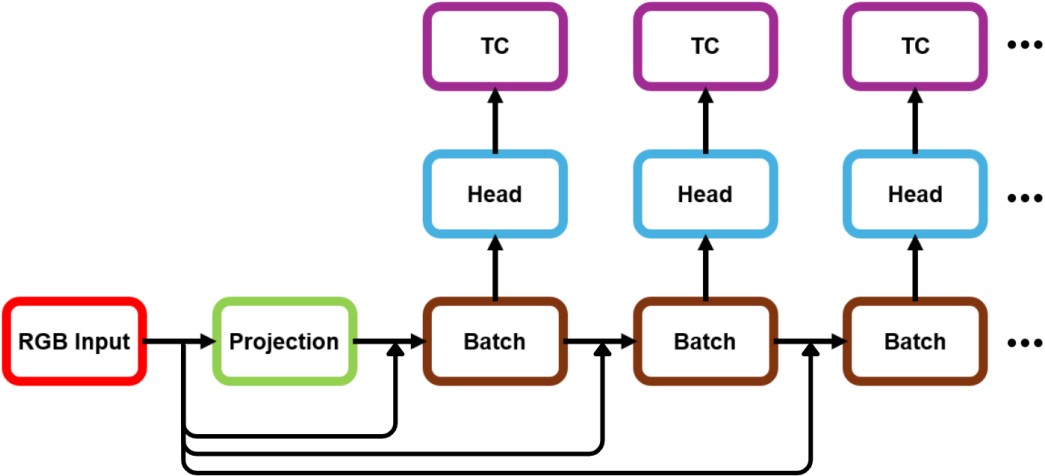

Figure 5: Block diagram for MazeNet.

To train this architecture, we generate a dataset of mazes with 2, 3, or 4 terminals, aiming to teach MazeNet to minimize the length when connecting a variable number of terminals. The exact method is computationally feasible for creating multiple optimal labels for low terminal mazes, allowing us to generate a large dataset for training. Refer to Section 4 for details on the training and testing data sizes. We use the progressive training strategy of Bansal et al. (2022), as it has demonstrated RCNN generalization capabilities when recurrence is applied. Progressive training encourages the model to incrementally refine its solutions from any given starting point. Specifically, we begin by inputting a problem instance and running the recurrent module for a random number of iterations, $n$. After this, we take the resulting intermediate output, reset the recurrence (discarding gradients from the initial steps), and then train the model to produce the final solution after an additional number of iterations, $k$. Here, $n$ is sampled uniformly from $[0, m]$, and $k$ is then defined after $n$ is randomly selected to fulfill the constraint $n + k = m$.

The output of the network, which matches the dimensions of the targeted image in Figure 2c and is a single-channel binary 0-1 valued matrix, is compared pixel-by-pixel with the labeled target using a cross-entropy loss function. The gradient calculation for progressive loss and backpropagation follows the procedure detailed in Algorithm 1 of Bansal et al. (2022).

Progressive training improves the quality of the network's incremental solutions, and by selecting a random iteration as the starting point for the next training phase, it also prevents the network from developing iteration-specific behaviors. Instead, the network is encouraged to learn iteration-agnostic behaviors.

We observe in Figure 6 that during the first ten epochs of training, MazeNet has a rapid increase in accuracy with each iteration. However, in the following ten epochs, the network exhibits diminishing returns (Appendix A.3).Importantly, we do not apply the termination condition during training,

which is why the accuracy does not reach 100% during training. Due to progressive training's inherent randomness, the model that is our definitive MazeNet is not necessarily the one from the final epoch, but instead the one with the highest peak accuracy on a test set with 5 terminals in each maze. Based on this criterion, we select the model from epoch 16 as the best-generalizing example of MazeNet in our implementation.

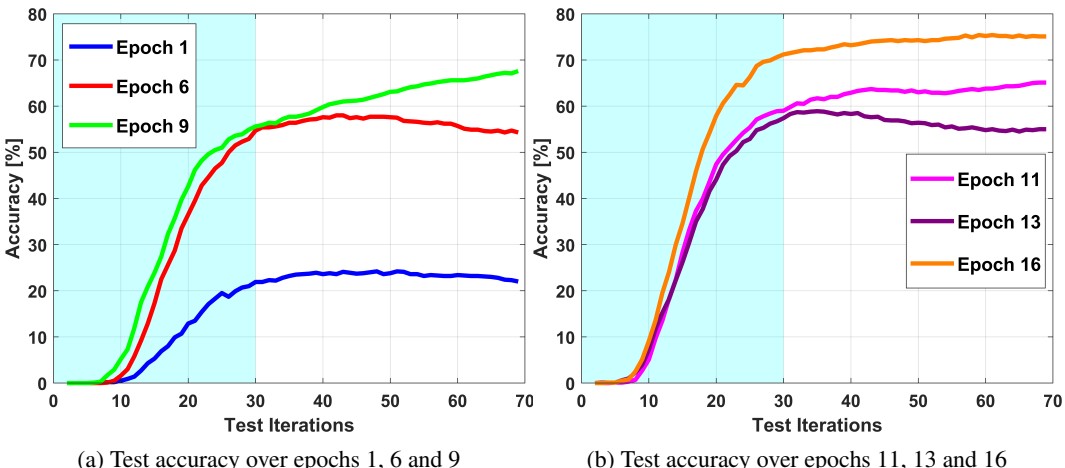

(a) Test accuracy over epochs 1, 6 and 9    (b) Test accuracy over epochs 11, 13 and 16

Figure 6: Accuracy progress for 5 terminals when training MazeNet, for several representative epochs for training regime $m = 30$ shaded in light blue in the background.

## 3.4 PARALLELIZATION FOR SCALABILITY

To address future scalability challenges, particularly in handling larger mazes that correspond to more complex graphs, we integrate multi-GPU parallelization to MazeNet directly at the algorithmic level. Unlike traditional multi-GPU approaches, where parallelization typically focuses on the gradient descent process, we parallelize the algorithms of MazeNet itself. This allows us to distribute the GPU convolution operations, while still maintaining the accuracy of the method.

In our approach, we divide the input image into smaller, equally-sized sections with each section processed independently through the projection, recurrent, and head modules in parallel, followed by a recombination of the sections at the end of this process. We divide the image into 2, 3, 4, 5, 6, 7, or 8 sections, as illustrated in detail in Appendix A.4. These image sections overlap by 10 pixels, which is sufficient to ensure that 2D convolution edge effects are discarded before the sections are combined, resulting in an output identical to what would be produced by processing the entire image as a single unit. Figure 7 also shows a specific example where two image sections are processed simultaneously through one iteration. The network is inherently parallelizable across all nine convolutional layers of a single feedforward pass without compromising any of MazeNet's accuracy. However, splitting and merging during every convolution introduces significant overhead. To minimize this overhead, we parallelize all the 9 convolution layers before merging the image sections.

For MazeNet's $48 \times 48$ images, the overhead introduced by parallelizing computations on a single GPU outweighs the performance gains due to the small image size. However, for larger synthetic images, such as $1000 \times 1000$, parallelization provides significant runtime improvements. The logic follows that convolution operations are inherently parallelizable without introducing errors. For larger synthetic images, parallelization gains runtime improvements in each iteration by efficiently distributing the computational load across sections.

## 4 RESULTS

We train MazeNet for 20 epochs with $m = 30$ iterations, using a $48 \times 48$ image size that represents graphs of $11 \times 11$ nodes. We select a width of 128 channels, which provides a sufficient number of parameters to maintain both accuracy and speed. We train with a dataset of 500,000 randomly

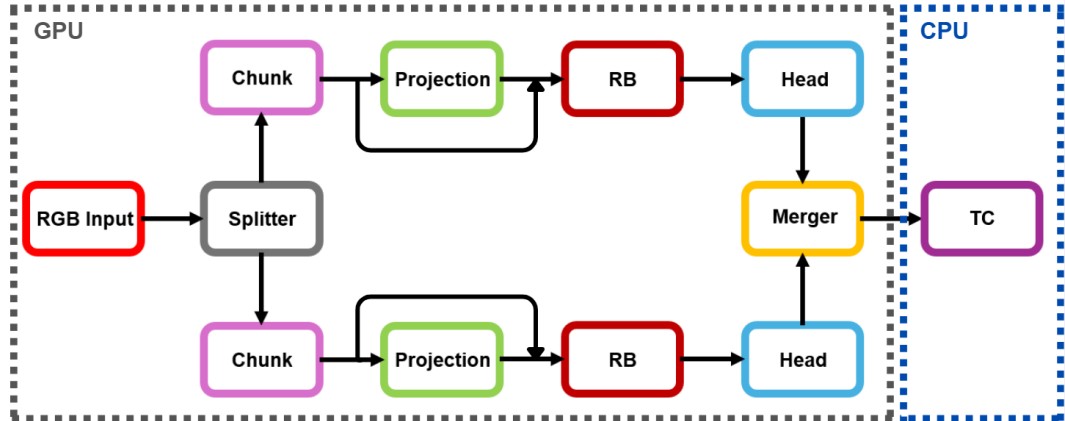

Figure 7: Block diagram of one parallelized iteration of MazeNet.

Table 1: Test accuracy (in %) with and without TC module over 20 MazeNet iterations.

| Number of Terminals | 2 | 3 | 4 | 5 | 6 | 7 |
|---|---|---|---|---|---|---|
| **TC module** | **100** | **100** | **100** | **100** | **100** | **100** |
| No TC module | 99.25 | 96.13 | 89.10 | 79.30 | 68.10 | 55.10 |

generated mazes with 2, 3, and 4 terminals. To evaluate MazeNet's generalization capabilities, we test on maze datasets containing 2 to 8 terminals. Each test set for 2 to 5 terminals has 10,000 mazes, but for 6 to 8 terminals we are limited to 1,000 mazes due high target generation runtimes. All of our experiments were conducted on NVIDIA GeForce GPUs, each with 11 GB of memory and a maximum power capacity of 250W. MazeNet was trained for approximately 48.12 hours across four GPUs, utilizing CUDA 11.4. In total, the training consumed 192.48 GPU-hours. This setup provided the computational resources necessary for parallel training of the model over 20 epochs. Further details are provided in Appendix A.6.

## 4.1 MAZENET ACCURACY AND PATH LENGTH COMPARISON ON TEST DATASET

MazeNet achieves 100% accuracy across the entire test set, even though it was trained only on a mixture of lower terminal numbers. We specifically define accuracy as the model's ability to identify the shortest path. As shown in Table 1, the TC module is critical for achieving such performance as the number of terminals grows. For cases without the TC module, the number of recurrent blocks must be arbitrarily defined regardless of any batch module; for comparison in Table 1, we set this number to 20 recurrent modules. As discussed before, approximation methods can be inaccurate for solving mazes; we provide the percentage accuracies for each method in Table 2.

In Table 3, we observe that the path length computed by MazeNet is always optimal, with differences between methods being small. This is because the majority of paths are identical across methods, and the average includes many such cases. However, when conditioned on MazeNet's mistakes, as shown in Table 4, the path length difference becomes more significant, ranging from 4

## 4.2 MAZENET RUNTIMES ON TEST DATASET

In Figure 8, we compare the runtime of MazeNet against Dijkstra's exhaustive Gutin & Punnen (2002), which is faster for a small number of terminals, but has excessive runtimes as the number of terminals increases. As expected, this brute force method scales linearly on a logarithmic scale, reflecting the factorial growth in complexity due to the permutation scaling. In contrast, MazeNet demonstrates much better scalability as the number of terminals increases. When compared with graph-based approximation methods (Kou et al., 1981), (Mehlhorn, 1988), we observe that the computational complexity for all three methods is comparable in terms of their scaling behavior. The runtime crossing point between Dijkstra's exhaustive algorithm and MazeNet occurs between 4 and

Table 2: Test accuracy (in %) for different maze-solving methods.

| Number of Terminals | 2 | 3 | 4 | 5 | 6 | 7 |
|---|---|---|---|---|---|---|
| **MazeNet** | **100** | **100** | **100** | **100** | **100** | **100** |
| Mehlhorn | 100 | 99.05 | 97.99 | 98.0 | 95.9 | 92.2 |
| Kou | 100 | 99.11 | 98.1 | 97.5 | 95.7 | 93.0 |

Table 3: Average path length normalized to Dijkstra's Exhaustive method across different maze-solving methods.

| Number of Terminals | 2 | 3 | 4 | 5 | 6 | 7 |
|---|---|---|---|---|---|---|
| **MazeNet** | **1.0** | **1.0** | **1.0** | **1.0** | **1.0** | **1.0** |
| Mehlhorn Method | 1.0 | 1.00002 | 1.00359 | 1.00871 | 1.01661 | 1.02037 |
| Kou Method | 1.0 | 1.000005 | 1.003559 | 1.008244 | 1.016068 | 1.021071 |

5 terminals, which is significant considering that we only trained MazeNet on a maximum of 4 terminals.

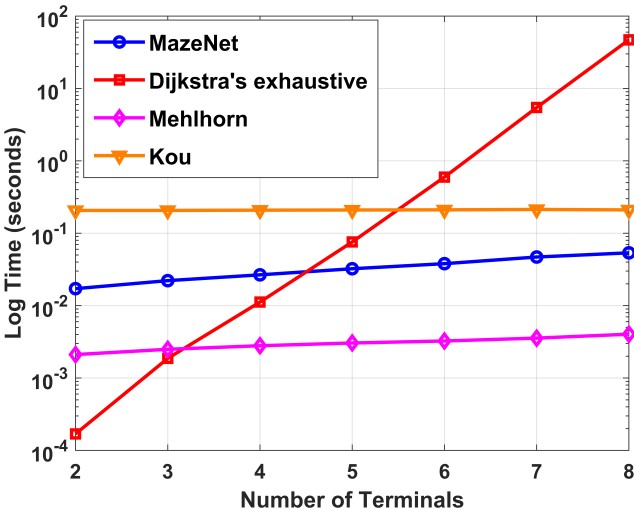

Figure 8: Mean runtime comparison between MazeNet, Dijkstra's exhaustive, and approximation methods.

The runtimes of Figure 8 demonstrate that, while Dijkstra's exhaustive algorithm is both accurate and efficient for low numbers of terminals, its runtime becomes prohibitive as the terminal number grows. In comparison, our method scales efficiently and remains practical even for larger numbers of terminals, offering a competitive runtime alternative to existing approximation methods, while also achieving the 100% accuracy that these methods cannot attain. An important achievement of MazeNet is that if we consider each pass through the recurrent module as one overall iteration, our method reaches the solution in very few iterations, as seen Appendix A.5, Figure 15. This contrasts with the competing methods, which often rely on loops that repeat for many more iterations to approximate a solution.

## 5 CONCLUSIONS AND FUTURE WORK

In this paper, we introduced the image-based MazeNet method for solving the Obstacle Avoiding Rectilinear Steiner Minimum Tree (OARSMT) problem, which achieves empirical accuracies of 100% and competitive runtimes compared to approximate methods that can produce incorrect results. MazeNet has promising generalization and scalability properties, but also offers valu-

Table 4: Average path length normalized to Dijkstra's Exhaustive method across different maze-solving methods (conditioned on approximations making a mistake).

| Terminals | 2 | 3 | 4 | 5 | 6 | 7 |
|---|---|---|---|---|---|---|
| **MazeNet** | **1.0** | **1.0** | **1.0** | **1.0** | **1.0** | **1.0** |
| Mehlhorn Method | 1.0 | 1.093595 | 1.061324 | 1.052873 | 1.055554 | 1.040212 |
| Kou Method | 1.0 | 1.094389 | 1.061881 | 1.053988 | 1.053699 | 1.041645 |

able insights into the relationship between image-based approaches and 2D graph structures. We demonstrated how an RCNN iteratively and successfully solves a challenging problem by emulating learned algorithmic steps, achieving state-of-the-art performance. Our method represents an integration of RCNNs with a search algorithm, opening new possibilities for hybrid approaches that combine the strengths of deep learning with traditional algorithmic techniques. Reframing graph problems as image processing tasks presents a promising avenue for leveraging recent advances in graph theory and in deep learning.

MazeNet establishes linkages between the fields of graph theory, deep learning, and traditional recursive algorithms. The insights to be gained extend far beyond the OARSMT problem, offering new directions for future research. For example, the intermediate steps of the algorithm that was learned by MazeNet do not resemble any known algorithmic solutions for the OARSMT problem. Therefore, closer study of this method could yield new classes of algorithms that differ substantially from the literature.

There are several important avenues for future work. First, while we successfully demonstrated that MazeNet can be generalized up to 8 terminals, the method's performance for larger terminal counts and larger mazes has not been explored yet. MazeNet can solve mazes with more terminals and arbitrary sizes without retraining; however, both convergence and optimality are no longer guaranteed under these conditions. Future research could focus on addressing these limitations to enhance the scalability and robustness of the method. As the dimensions of the mazes grow, simply training the current MazeNet on such augmented datasets may become computationally prohibitive, and additional preprocessing steps may be required. Also, an adaptation of MazeNet to layouts beyond mazes, such as those used in (Chen et al., 2022), (Lin et al., 2023), (Huang & Young, 2011), and (Chu & Wong, 2008), would allow for a more direct comparison to these approaches. Second, MazeNet is a deep learning approach, which is being compared to graph algorithms. It is a difficult comparison given that both approaches are dissimilar. Graph Neural Networks (GNNs) (Micheli, 2009) present a deep learning alternative that is highly comparable to MazeNet. GNNs are specifically designed for tasks involving graph-structured data, such as the OARSMT, which is inherently a graph problem. Recent developments in GNNs have sparked significant interest, further solidifying their utility in tasks with close ties to our dual approach involving graphs and image-based computer vision tasks (Han et al., 2022). Future research on GNNs inspired by MazeNet could complement the recurrent approach proposed in this work, potentially leading to insightful and significant findings. Lastly, adapting MazeNet's architecture and training regime to application-specific instances of the OARSMT problem could provide a more specialized solution through domain-specific labeled paired examples.

We demonstrate the high accuracy, fast runtimes and scalabilty of MazeNet, showing the potential of hybrid approaches that blend deep learning with algorithmic techniques. MazeNet offers a foundation for exploring not only other graph-related problems but also more general optimization tasks. As deep learning continues to advance, methods like MazeNet could play a key role in shaping how complex combinatorial problems are tackled in future research.

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

# A APPENDIX

## A.1 DATA GENERATION

We begin with a grid graph where each node is connected to four neighbors—two vertical and two horizontal—except at the boundaries. From this uniform structure, we generate a perfect maze using the DFS algorithm, a widely recognized method for maze construction. While DFS is well-documented in the literature, we detail the process here for our specific method.

In this approach, the maze is represented as a grid of cells, each initially enclosed by four walls: North, East, South, and West. The algorithm starts from an arbitrary cell and inspects its neighboring cells. If any of the neighbors have not been visited, the algorithm randomly selects one, removes the wall between the two cells, and moves to the unvisited cell. This process is repeated, continuously extending the path. If the algorithm encounters a dead end, where all neighboring cells have already been visited, it backtracks to the last cell with unvisited neighbors and continues from there.

In our implementation, the DFS algorithm is managed by defining classes for both the individual cells and the overall maze structure. We track the path through the grid using a stack, which is implemented as a Python list. When the algorithm encounters a dead end, it backtracks by popping cells off the stack until it finds a cell with unvisited neighbors, allowing the exploration to continue.

This process results in a perfect maze—one in which each cell is reachable through a single connected path, and in which no cycles exist. To increase the complexity and to introduce cycles, we further modify the maze by randomly removing walls. Priority is given to nodes that are dead ends, i.e., cells with three remaining walls, in order to break up isolated paths and to add variability. This introduces cycles into the maze, making it more complex and extending MazeNet to a wider class of problems.

Finally, after constructing the maze, we uniformly select $N$ terminals at random locations throughout the grid. These terminals are the points that will be connected in our OARSMT solution as seen in Figure 9. By introducing these random elements, we ensure a diverse and challenging dataset for our purposes.

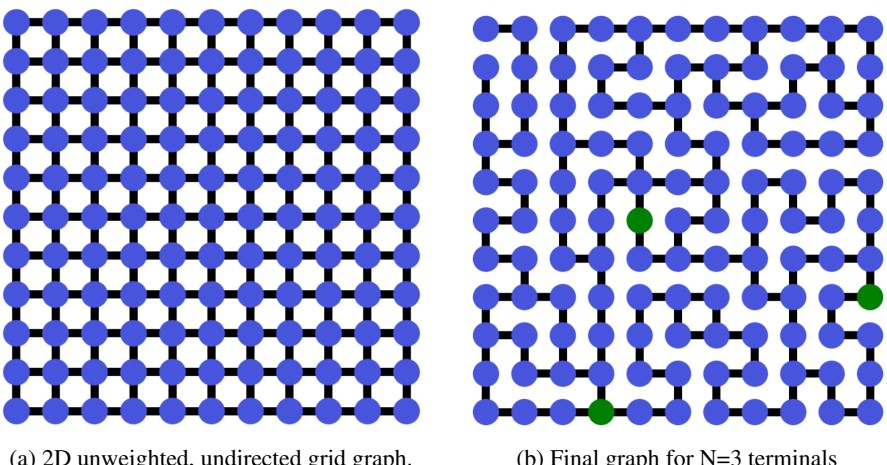

(a) 2D unweighted, undirected grid graph.          (b) Final graph for N=3 terminals

Figure 9: An example of the process of graph generation.

Once created, the graph can be converted to a maze. We create the target (i.e., solution) with Dijkstra's exhaustive, having the labeled pair as shown in Figure 10.

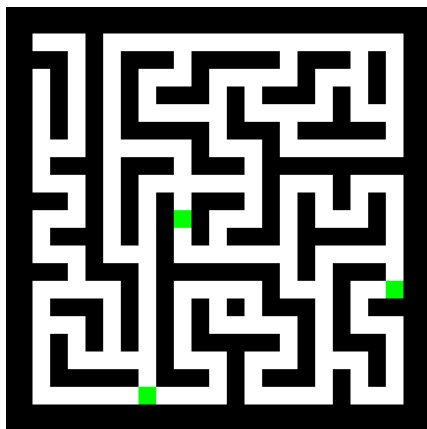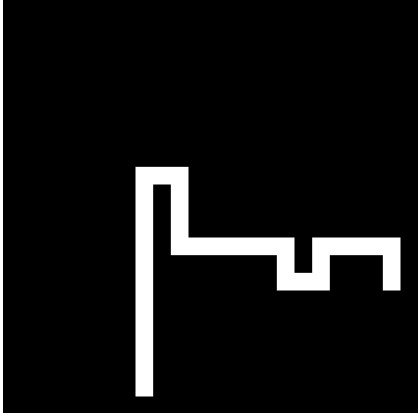

Figure 10: Labeled-pair example for 3 terminals.

## A.2 DETAILS OF MAZENET'S ARCHITECTURE

As depicted in Figure 11, MazeNet's architecture consists of nine convolutional layers. The input image, which is 3-channel and of arbitrary size ($48 \times 48$ in our case), first passes through the projection module. This module is a single convolutional layer (Conv 0) that expands the input image to a predefined number of width channels (set to 128 in this project) while maintaining the spatial dimensions of $48 \times 48$. A ReLU activation follows this layer, introducing non-linearity and allowing for more expressive feature extraction.

The core of the architecture is the recurrent module (RB), which plays an essential role in the model's ability to generalize. The recurrent module consists of five convolutional layers (Conv 1-5). The first layer in the module takes a concatenation of the input channels and the width channels from the previous iteration (input + width channels), ensuring that information is retained throughout the iterations. It outputs the same number of width channels (128 in this case). The remaining four layers are identical convolutional layers, processing and outputting width channels without additional activations. This recurrent structure is applied iteratively during the testing phase, allowing the model to adapt and generalize across different terminals.

The final component is the head module, which is responsible for reducing the channel width and producing the (0-1) output. It achieves this by progressively decreasing the number of channels through three convolutional layers (Conv 6-8) with ReLU activations between them. The first layer reduces the channels from 128 to 32, the second reduces it further from 32 to 8, and the final convolution narrows the channels from 8 to 2. An Argmax function is applied over the two channels, generating a binary prediction (0-1) for each pixel. For the final model in our architecture in Figure 5, we use the Batch modules as as detailed in Figure 12.

## A.3 DETAILS OF THE TRAINING PROCEDURE

To evaluate performance during the training process, we plot the mean iteration accuracy for each epoch, starting from iteration 2 through iteration 70 — more than double the size of the training regime. Importantly, we do not apply the termination condition during training. This decision is made to keep all operations confined to the GPU and to accelerate the training process, as the termination condition introduces additional overhead. By calculating accuracy without the termination condition, we ensure faster evaluations while maintaining GPU efficiency.

For model selection, we choose the model based on the highest peak mean accuracy in Figure 13, irrespective of the iteration number. This method allows us to identify the best-performing model without overemphasis on any specific iteration. The accuracy curve shows that the model at many

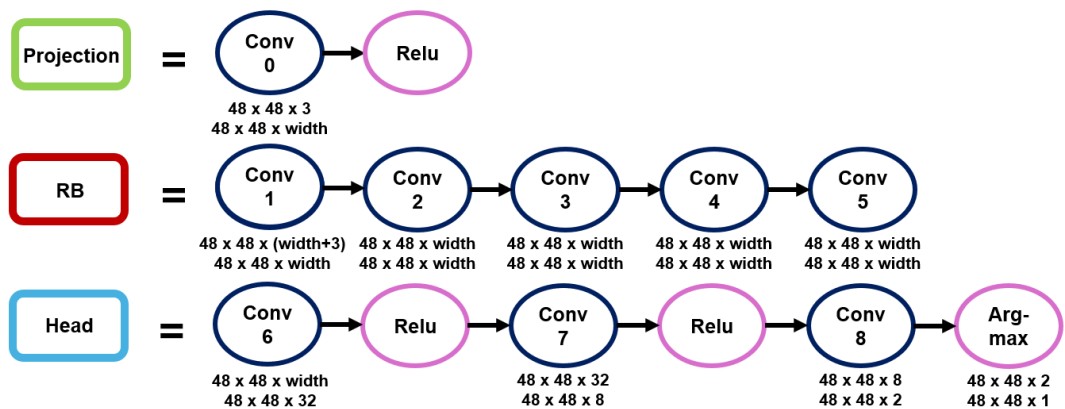

Figure 11: Implementation details of the Recurrent, Projection and Head blocks of MazeNet.

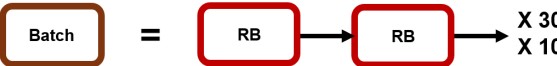

Figure 12: The first Batch module in Figure 5 is composed of 30 RB modules in total, while the extra recurrent Batch modules are 10 RB modules in total.

epochs generalizes effectively outside the training regime, as accuracy consistently increases beyond the maximum iterations used in training.

### A.4 PARALLELIZATION RUNTIME

For MazeNet's $48 \times 48$ images, we have found that the overhead introduced by parallelizing the computations on a single GPU outweighs the performance gains, due to the relatively small image size. However, when experimenting with larger synthetic images—such as of dimensions $1000 \times 1000$—we can obtain significant runtime improvements. To quantify these runtimes, we split up such images into a variable number of sections, from 1 (no parallelization) to 8 (maximum parallelization on the GPU), and pass them through the network in parallel. We obtain the runtime results in Figure 14, where the maximum runtime savings are obtained for 2 sections, and the parallelization overheads leading to smaller savings for increasing numbers of sections.

### A.5 NUMBER OF ITERATIONS

An important achievement of MazeNet is that if we consider each pass through the recurrent module as one overall iteration, our method reaches the solution in very few iterations, as seen in Figure 15. This contrasts with the competing methods, which often rely on loops that repeat for many more iterations to arrive to a solution. Our method's rapid convergence reduces the number of required iterations, minimizing computation time while maintaining 100% accuracy.

### A.6 HYPERPARAMETERS

To ensure that the runtime reflects the actual time required to solve each maze, we use a test batch size of 1. Larger batch sizes would introduce GPU parallelism during the convolutions, resulting in artificially lower runtimes, which would skew the comparison with graph techniques. By maintaining a batch size of 1, we provide a fair and accurate assessment of the time required to solve each maze individually. The $\alpha$ parameter is the parameter used in Schwarzschild et al. (2021), Algorithm 1 for progressive training. The full list is displayed in Table 5

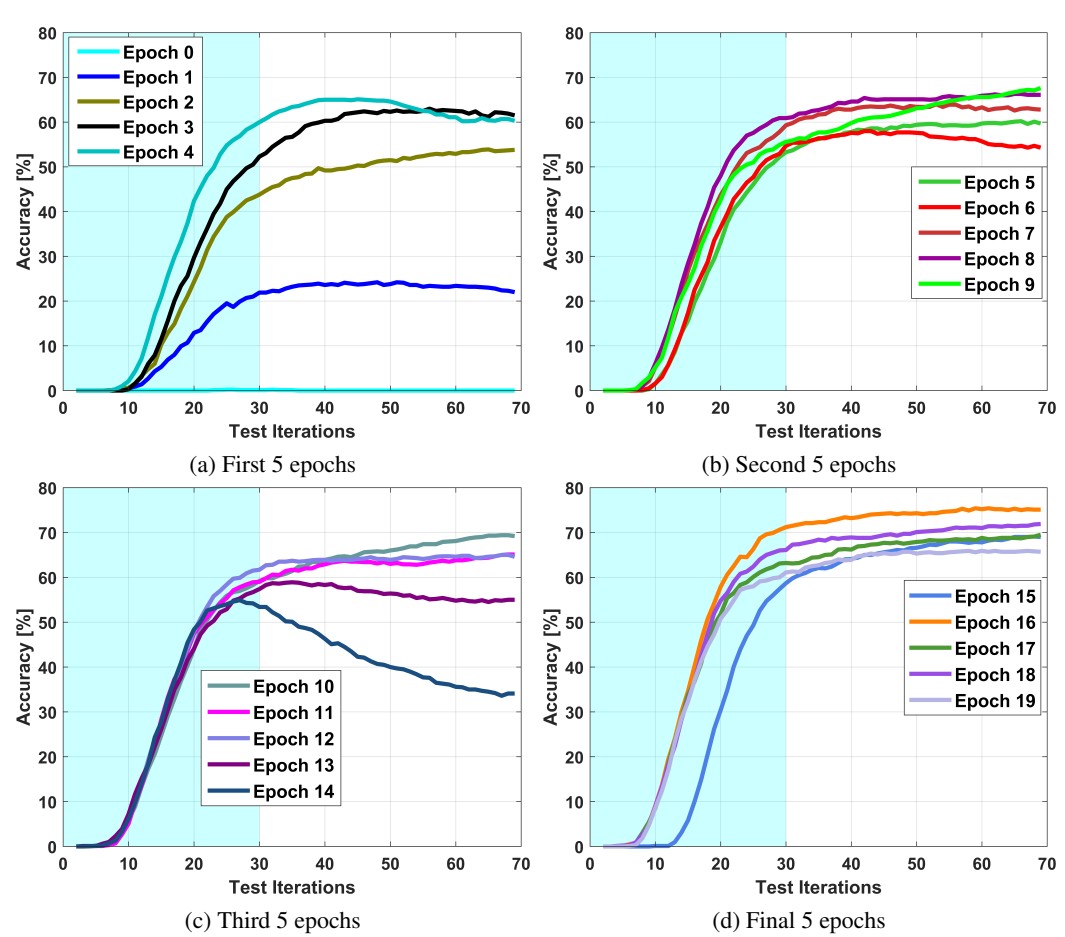

Figure 13: Test accuracy progress for 5 terminals when training MazeNet, for several representative epochs for training regime $m = 30$ shaded in light blue in the background.

Table 5: Training and Testing Parameters

| Parameter | Value |
| --- | --- |
| $\alpha$ | 0.01 |
| Epochs | 20 |
| Learning Rate | 0.001 |
| Optimizer | Adam |
| Test Batch Size | 1 |
| Train Batch Size | 25 |
| Train Mode | Progressive |
| Width | 128 |
| Training Regime (m) | 30 |
| Test Iterations (Low) | 2 |
| Test Iterations (High) | 70 |
| Test Data Type | 5_green |

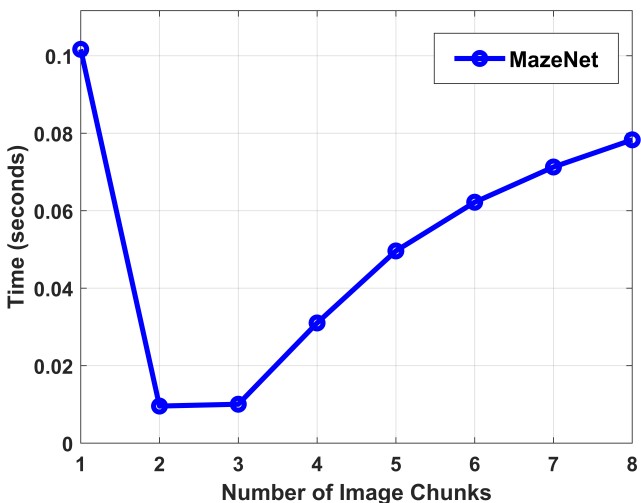

Figure 14: Parallelized MazeNet's runtimes for a single iteration through the network, as a function of number of sections, for images of dimensions $1000 \times 1000$.

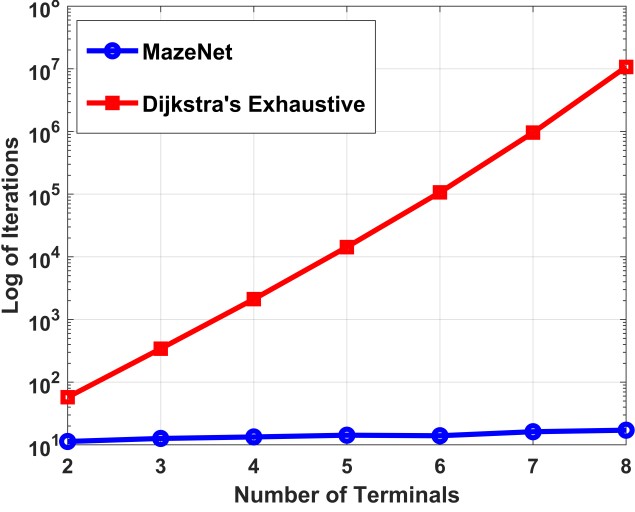

Figure 15: Mean iteration comparison between MazeNet and Dijkstra's exhaustive method.

