# OpenReview forum: "MazeNet: An Accurate, Fast, & Scalable Deep Learning Solution for Steiner Minimum Trees"
_ICLR.cc/2025/Conference — Submitted to ICLR 2025_

### Official Review · Reviewer_psAX · 2024-10-22

**Soundness:** 2
**Presentation:** 2
**Contribution:** 2
**Rating:** 3
**Confidence:** 4

**Summary:**

This paper addresses the Obstacle Avoiding Rectilinear Steiner Minimum Tree (OARSMT), which seeks to find a set of horizontal and vertical connections between a set of points while avoiding obstacles using the minimum overall connection length. The paper's technical approach is to convert OARSMT graphs to images then use a Recurrent Convolutional Neural Network (RCNN) to iteratively highlight the solution. RCNN-based solutions to OARSMT were introduced in previous work, but this paper uniquely extends RCNN-based maze solving to larger maze domains with more terminals where traditional methods are computationally inefficient. In addition, this paper develops a termination condition to avoid both premature termination and excessive runtimes. Finally, this paper includes experimental results with 2-7 terminals in 11x11 mazes with 100% accuracy.

**Strengths:**

Approach for converting Obstacle Avoiding Rectilinear Steiner Minimum Tree (OARSMT) problem to image-based Recurrent Convolutional Neural Network (RCNN) with extensible training images and more than 2 terminals.

100% empirical accuracy on test cases (40,000 total mazes for 2-5 terminals and 3,000 mazes for 6-8 terminals). Alternatively, graph-based approximation methods of Kou et al. 1981 and Mehlhorn 1988 have errors with 3 or more terminals.

MazeNet is computationally faster than Dijkstra's algorithm when 5 or more terminals are used.

Maze figures are straightforward and informative (e.g., Figure 4).

**Weaknesses:**

Several details technical details are unclear (see specific feedback below).

Does not provide any limitations or failure cases. For example, what happens if >> 8 terminals are used? This is only discussed as future work. Does algorithm run indefinately for unreachable terminals?

A lot of overlap with Schwarzschild et. al. 2021, but with additional terminals and the terminal condition module.

The paper emphasizes that their approach is parallelizable (L23, L155, L315) but does not provide key details on how this approach works or report accuracy of experimental results on larger mazes to verify it's utility. Instead, the paper provides a vague description of the parallelization process (Section 3.4, L315) and reports only on runtime performance from parallelization on larger mazes (Figure 9, L466).

**Questions:**

## Questions

How would researchers replicate your work?

L111 How is O(T!) permutations determined for exhaustive methods?

What is the purpose of the paragraph at L174-182? Is the progressive training algorithm of Bansal et al. used in this work? If so, be explicit and state that.

At L224, "...position, indicating a cycle, it is terminated to prevent redundant processing." After finding a cycle and terminating, which single path is chosen?

Algorithm 1 L245-250 is a bit difficult to follow. "junction found" can only be understood by referencing back to the text. Also, what if the "Move to the direction with highest 'whiteness'" is in the backwards direction?

L269 Why are mazes of 2, 3, or 4 terminals chosen for training? (e.g., as opposed to 5, 6, 7)

L293-295 reference random variables n,k. What distributions are these sampled from?

Parallelization for Scalability Section 3.4 is missing specific details.
How many sections are images divided into? (L320)
How many pixels are "sufficient" overlap? (L322)
For a section with two or more terminals, what is the incentive to find additional paths to other unknown sections?
What is the goal of a section with only one terminal?
How does parallelization work for sections without terminals?

L378 What does "20 MazeNet iterations" refer to? Earlier sections indicated that 30 module iterations are used before checking terminal conditions (L261) and 16 training epochs are used (L310). There is no explanation in the text or table.

## Feedback

L55 describes a 11x11 maze, but the paper does not clarify what "11" refers to until L125 in Section 2.1. Explain what 11x11 means at L55 (e.g., "11x11 node graph").

Figure 5 is first referenced at L266 but provides almost no detail or context for what the "Projection," "Batch," and "Head" blocks are. Projection was referenced once at L176 when discussing another paper's work. Multiple configurations of the batch and head modules are referenced earlier, but all blocks are uniformly labeled without any specification of the differences between them. For example, the first "Batch" represents 30 RB iterations and subsequent "Batch" represents 10 iteration (L261) but these are labeled as the exact same module in Figure 5. As another example, L177-180 reference a "Head" module that produces the output and a "final head module" that transforms the network's output to single-channel prediction. Why not add these details to Figure 5 to be more informative and accurate?

---

> ### Author Response · Authors · 2024-11-27
> **Responses to Reviewer psAx Comments**
>
> 1. "Limitations and Failure Cases with More Than 8 Terminals": Thank you for highlighting the importance of addressing MazeNet’s limitations in scenarios with significantly more than 8 terminals. We acknowledge that MazeNet does not guarantee convergence or optimal path length in such cases. To manage these limitations, MazeNet incorporates a predefined maximum number of iterations. If a solution fails to converge or terminals remain unreachable, the algorithm halts and marks the attempt as unsuccessful. These constraints will be explicitly stated in the revised manuscript to ensure clarity.
>
> 2. "Parallelization Process and Figure 14 Details": We appreciate the request for more specific details on the parallelization process. In Figure 14, the x-axis represents the number of image chunks into which the input is divided, with a 10-pixel overlap between chunks. This overlap ensures no errors are introduced at chunk boundaries, considering the kernel size for convolutions is 3. If the overlap is smaller than the number of convolutions per iteration, errors accumulate at chunk boundaries. By maintaining a 10-pixel overlap, we ensure equivalence to performing a convolution on the entire image, regardless of how terminals are distributed across image sections. These details will be added to Section 3.4 to provide a comprehensive explanation of the parallelization process. We will include these specific details in the revised manuscript to provide a complete explanation. Thank you for highlighting this aspect, as it has allowed us to improve the clarity of this section.
>
> 3. "Replicability and Code Release": We deeply value the reviewer’s emphasis on replicability. To facilitate this, we are preparing the MazeNet codebase for public release. The released version will include detailed documentation, implementation instructions, and explanations to enable researchers to replicate our work accurately. Furthermore, we will elaborate on the T! complexity, which arises from permuting terminals and sequentially connecting them in pairs, ensuring all possible orderings are considered.
>
> 4. "Progressive Training Algorithm and Cycle Detection": The paragraph in lines 174–182 refers to the progressive training algorithm of Bansal et al., which is explicitly employed in this work. While our intention was to provide context for our approach, we appreciate the reviewer’s suggestion for greater clarity. In the revised text, we will explicitly state our use of their algorithm to ensure understanding.
>
> 5. "For cycle detection": We thank the reviewer for bringing attention to this point. The TC module discards solutions when cycles are detected, requiring additional iterations to clear the incorrect solution. We will clarify this process further in the revised manuscript.
>
> 6. "Algorithm 1": We are grateful for the opportunity to refine the explanation of Algorithm 1. The algorithm explicitly prohibits retracing steps by storing the last move. We agree that Algorithm 1 requires additional clarity, and we will explicitly redefine the condition "junction found" for consistency. Additionally, we will provide further detail on how the direction with the highest "whiteness" is selected, and emphasize that backward moves are explicitly disallowed in the revised version.
>
> 7. "Training configurations": We appreciate the reviewer’s insights regarding training maze configurations. To balance complexity and simplicity, we selected setups with 2, 3, or 4 terminals. Configurations with fewer terminals are less computationally expensive to generate using exact optimal methods, enabling us to create a high-quality training dataset of 0.5 million examples. As the number of terminals increases, the time required per maze becomes significant, making larger configurations impractical at this scale. We will ensure this reasoning is clearly articulated in the revised manuscript.
>
> 8. "Random Variables":  We thank the reviewer for their observation regarding the random variable n. The random variable n, mentioned in lines 293–295, is sampled from a uniform distribution. We will explicitly include this detail in the revised manuscript for clarity. Once n is determined, k is also determined so that it satisfies the constraint n+k=m.
>
> 9. "RB Module Iterations and Scaling Limitations": We appreciate the reviewer’s observation regarding the 20 MazeNet iterations referenced in line 378. These iterations correspond to the number of RB module iterations used for comparison with the TC module. Without the TC module, the RB module runs for a predefined number of iterations, which we set to 20 in Table 1. We will clarify this in the revised text.

---

### Official Review · Reviewer_urst · 2024-11-03

**Soundness:** 2
**Presentation:** 2
**Contribution:** 2
**Rating:** 3
**Confidence:** 4

**Summary:**

The article establishes a MazeNet model to solve the OARSMT problem. Specifically, it first converts the graph representation of the maze into image representation, then processes the image data using the RCNN model, and finally reduces the model's running time through a termination condition.

**Strengths:**

The application is interesting.

**Weaknesses:**

1.	The motivation is not clear, as the article does not explicitly outline the problems with previous solutions to the OARSMT problem, nor does it explain how this article addresses these issues.
2.	The experimental evaluation metric design is unreasonable. The OARSMT problem is an NP-hard problem. However, the evaluation metric used in this article's experimental section is accuracy. While for small-scale problems, the shortest path can be obtained using Dijkstra's algorithm for comparison to calculate precision, for large-scale problems, it is challenging to solve using Dijkstra's algorithm.
Furthermore, the second part of the article clearly states that the optimization goal is to minimize path length. However, the evaluation metric in the experimental section does not use path length as a measure, which is confusing.
3.	In line 164 of the text, it is stated that "However, these problems were in domains where traditional methods are both fast and accurate, leaving open the question of whether RCNNs can provide similar advantages for more complex graph-based problems." Given that traditional algorithms can achieve good results, what is the significance of this research? Moreover, the question of whether RCNNs can provide similar advantages for more complex graph-based problems remains unresolved. How does this study address or prove this issue?
4.	The resolution of figures 2b and 2c is too low. Although the generated data size is 48x48, clear images should still be placed in the article.
5.	The author's proficiency in English is lacking, and the translation traces are too obvious.
The innovation in this article is weak. Regardless of whether it is RCNN or the conversion of graph representation to image representation, the innovation is very limited. From both a writing and experimental perspective, it resembles more of an experimental report and is not suitable for publication as a research paper.

**Questions:**

1. The article only mentions the number of samples in the test set. What is the number of samples in the training set?
2. In terms of problem scale, for instance in the field of chip design where there are tens of thousands of nodes with connections that must adhere to certain constraints, can this algorithm achieve good results in larger-scale tasks?
3. The testing accuracy can reach 100%, could this be a result of overfitting?

---

> ### Author Response · Authors · 2024-11-27
> **Responses to Reviewer urst Comments**
>
> 1. "Motivation and Limitations of Previous Solutions to the OARSMT Problem": Thank you for emphasizing the need for clearer motivation. In the revised manuscript, we have explicitly highlighted the motivation behind MazeNet. This work addresses the gap between probabilistically correct approximation methods and the demand for deterministic accuracy in small-scale environments. For problems within the 11×11 regime, probabilistic methods may be effective, but they lack justification for failing to discover the true shortest path. MazeNet is designed to achieve deterministic accuracy comparable to exact graph-based methods while maintaining the runtime efficiency of approximation algorithms. This approach fills the gap for small but complex maze-like problems where suboptimal paths are unacceptable. Additionally, we discuss MazeNet's ability to generalize beyond this specific regime, while noting that scaling to larger mazes is an interesting avenue for future research, albeit outside the scope of this work.
>
> 2. "Evaluation Metric and Experimental Design": We appreciate the reviewer's insight regarding the evaluation metric and experimental design. Accuracy, as used in our evaluation, reflects the ability to determine whether the shortest path is found, particularly in small-scale mazes where exact solutions can be determined using methods like Dijkstra's exhaustive algorithm. While this may not be a conventional metric for OARSMT problems, it aligns with our goal of ensuring deterministic correctness in this regime. To complement this, we have now included a wirelength ratio comparison to provide additional context. Furthermore, we present the average wirelength percentage increase conditioned on approximation errors. These results highlight the operational significance of errors in suboptimal paths and demonstrate how MazeNet, achieving 100% accuracy, avoids this issue entirely.
>
> 3. "Significance of This Research and Comparison with RCNNs": The significance of this work lies in demonstrating the capability of RCNNs to solve more complex graph-based problems through the incorporation of novel modules such as the Termination Condition (TC) module and reorganized recurrent blocks in a batch module. These enhancements enable MazeNet to achieve deterministic accuracy and runtime efficiency for small maze-like problems, addressing the trade-off between accuracy and speed that is seen in competing methods. As shown in Tables 1, 2, and Figure 8, MazeNet achieves optimality and practicality within the 11×11 regime, handling up to 8 terminals. This experimentally demonstrates that RCNNs can address complex graph-based problems in this setting, contrasting with prior RCNN applications, which have been limited to simpler tasks like connecting two nodes in a maze.
>
> 4. "Figures 2b and 2c Resolution": We acknowledge the issue with the resolution of Figures 2b and 2c. While the underlying data is represented in a 48×48 format, we will update these figures with higher-resolution versions in the revised manuscript to improve their clarity and visual quality.
>
> 5. "English Proficiency and Writing Style": We apologize for any lack of clarity or instances where the writing appeared to have traces of translation. We appreciate the reviewer’s feedback on this matter and are committed to improving the quality of the manuscript. If the reviewer could point to specific examples or sections where the language could be refined, we would be grateful and will address these issues in the revision.

---

### Official Review · Reviewer_ioWM · 2024-11-03

**Soundness:** 3
**Presentation:** 3
**Contribution:** 3
**Rating:** 5
**Confidence:** 4

**Summary:**

MazeNet, a recurrent convolutional neural network (RCNN) for the Obstacle Avoiding Rectilinear Steiner Minimum Tree (OARSMT) problem, shows promise with 100% accuracy in initial tests but requires further validation on larger grids and more terminals to confirm scalability. Questions remain on its novelty, given similar RCNN applications in maze-solving, and on its high training time (48.12 hours on four GPUs), along with the need to reduce training data complexity and evaluate the TC module's computational overhead. Additional context through a more detailed literature review would also strengthen the work.

**Strengths:**

1. MazeNet is designed for scalability and adaptability, making it effective for solving mazes of varying sizes and numbers of terminals that need connection.

2. While RCNNs alone may struggle to identify and verify a correct solution to terminate the process, MazeNet addresses this by incorporating a search-based algorithm that reliably detects a correct solution. This approach combines the speed of graph-based approximate algorithms with the precision of exhaustive graph-based methods.

3. RCNNs provide step-by-step interpretability of the method’s operations, as the head module can be applied at any iteration, allowing for observation of intermediate solution stages. These stages can be visualized as image outputs, providing insight into the solution process at each step.

**Weaknesses:**

1. The proposed approach of using a recurrent convolutional neural network (RCNN) to solve the Obstacle Avoiding Rectilinear Steiner Minimum Tree (OARSMT) problem may lack novelty, as RCNNs have previously been applied to similar maze-solving problems.

2. Although MazeNet demonstrated 100% accuracy in the reported experiments, additional proof is needed to confirm it can consistently achieve this level of accuracy across all problem instances.

3. The experimental setup appears limited; testing just on a grid of 11 × 11 nodes with up to 8 terminals may not be sufficient to thoroughly assess MazeNet’s performance, particularly regarding its scalability.

4. While the TC module improves MazeNet's accuracy, it introduces significant computational overhead, which has not yet been systematically evaluated.

5. The paper lacks a dedicated related work section, and a more comprehensive discussion of relevant literature would strengthen the context for this research.

**Questions:**

1. In what ways does the proposed method differ from prior work that applies Recurrent Convolutional Neural Networks (RCNNs) to solve maze-related problems?

2. Does MazeNet require separate training for different grid and terminal configurations, such as an 11×11 versus a 9×9 node grid, or can a single model handle multiple setups?

3. What strategies can be employed to reduce the time and computational complexity involved in generating training data?

4. Training MazeNet reportedly took around 48.12 hours across four GPUs, which is considerable. How does training time scale with increased problem complexity and size, and what optimizations could help reduce this duration?

5. In Figure 8, is the runtime of MazeNet measured with parallelization applied?

---

> ### Author Response · Authors · 2024-11-27
> **Responses to Reviewer ioWM Comments**
>
> 1. "How does the proposed method differ from prior RCNN-based approaches for maze problems?": Thank you for highlighting this important question. Previous works using Recurrent Convolutional Neural Networks (RCNNs) have largely focused on simpler maze-related problems, such as connecting two nodes. These problems often have efficient optimal algorithmic solutions, making neural network-based approaches less compelling for such tasks. While these prior works are valuable for their exploration of architecture and extrapolation pipelines, the problems addressed remain insufficiently complex to advance beyond state-of-the-art methods. In contrast, our work applies RCNNs to an actual NP-hard problem. We extend existing architectures by introducing a novel Termination Condition (TC) module and reorganizing batch modules, enabling the model to solve small but relevant maze-like problems optimally. These contributions address a gap not explored in prior RCNN-based research.
>
> 2. "Does MazeNet require separate training for different grid and terminal configurations?": MazeNet’s architecture is designed to handle arbitrary maze sizes and terminal configurations without requiring separate training for each setup. However, in the absence of experimental guarantees for larger mazes or configurations, we acknowledge its potential limitations compared to approximation methods when scaling beyond the demonstrated 11×11 regime. Our focus in this paper is on achieving deterministic optimality in small mazes, where suboptimal solutions are unacceptable. While the architecture can generalize to larger mazes, this is outside the scope of the current work, and we aim to explore these possibilities in future research.
>
> 3. "What strategies can reduce the time and computational complexity of generating training data?": We are currently exploring the use of exact solvers to generate training labels, as these ensure the model learns to consistently find optimal solutions. Approximation methods, while computationally faster, are unsuitable for this purpose as they cannot guarantee optimality. Ensuring the correctness of training labels is a priority, even at the expense of additional computational effort. Moving forward, we aim to optimize the solver to accelerate training data generation while maintaining label accuracy.
>
> 4. "How does training time scale with increased problem complexity, and what optimizations could reduce this duration?": In our experiments, increasing maze size did not significantly affect training time. For the 11×11 regime, the training duration (~48.12 hours across four GPUs) was sufficient for the intended application. While this work does not focus on reducing training duration, we are actively investigating optimizations to scale the process efficiently for larger problems.
>
> 5. "Runtime Measurement in Figure 8": We appreciate the opportunity to clarify this point. The runtime reported in Figure 8 corresponds to executions without parallelization for smaller 48×48 images, as we observed that the overhead introduced by parallelization outweighed any potential benefits in these cases. However, for synthetic 1k×1k images (~500×500 node graphs), parallelization demonstrated significant runtime improvements, which is why those results are highlighted in Figure 14. We hope this clarification provides a clearer understanding of the context and focus of our runtime analysis.

---

### Official Review · Reviewer_7pkf · 2024-11-04

**Soundness:** 2
**Presentation:** 3
**Contribution:** 2
**Rating:** 3
**Confidence:** 3

**Summary:**

This paper proposes MazeNet, a learning-based algorithm that leverages a recurrent convolutional neural network to predict a single-channel binary matrix iteratively, thereby solving the Obstacle Avoiding Rectilinear Steiner Minimum Tree (OARSMT) problem. The algorithm is evaluated on different mazes with 2-8 terminals, showing 100% test accuracy and competitive planning speed.

**Strengths:**

1. This paper formulates the OARSMT into a binary image prediction problem, which is easy to understand and reasonable.

2. The experimental results show that MazeNet is able to achieve an impressive 100% test accuracy.

3. The experimental results show that MazeNet scales well with an increasing number of terminals.

**Weaknesses:**

1. The mazes that MazeNet is evaluated on are too small, of only 11 x 11 kernels. There is not strong evidence that MazeNet can perform well on larger mazes.

2. This work only compares MazeNet with classical solvers like Dijkstra, Mehlhorn and Kou, etc. However, there are some more recent algorithms that are either learning-based or CPU-based, e.g., [1], [2]. [3]. Comparison with more and stronger baselines is needed to consolidate the conclusion.

3. It is not new to learn to predict the future images, e.g., [4] also formulated the grid-like motion planning problem into a video prediction problem. From this paper, I can not see how the specific domain knowledge from OARSMT is incorporated into the network design.


[1] Lin, Zhenkun, et al. "Obstacle-Avoiding Rectilinear Steiner Minimal Tree Algorithm Based on Deep Reinforcement Learning." 2023 International Conference on Artificial Intelligence of Things and Systems (AIoTSys). IEEE, 2023.

[2] Chen, Po-Yan, et al. "A reinforcement learning agent for obstacle-avoiding rectilinear steiner tree construction." Proceedings of the 2022 international symposium on physical design. 2022.

[3] Huang, Tao, and Evangeline FY Young. "An exact algorithm for the construction of rectilinear Steiner minimum trees among complex obstacles." Proceedings of the 48th Design Automation Conference. 2011.

[4] Zang, Xiao, et al. "Robot motion planning as video prediction: A spatio-temporal neural network-based motion planner." 2022 IEEE/RSJ International Conference on Intelligent Robots and Systems (IROS). IEEE, 2022.

**Questions:**

1. How is the threshold 0.65 decided as the TC threshold? Is there ablation study to find the optimal value?
2. What is the step size of the solver, i.e., how many cells are the trees extended in each iteration? How many one entries are contained in the predicted binary matrix?
3. Curious what is the performance of MazeNet on large mazes, e.g., 256 x 256?

---

> ### Author Response · Authors · 2024-11-27
> **Responses to Reviewer 7pkf Comments**
>
> 1. "Evaluation on Small Mazes (11x11)": Thank you for pointing out the potential limitations of evaluating MazeNet on small 11x11 mazes. Our motivation for MazeNet, which we acknowledge was not sufficiently clarified in the original manuscript, is rooted in the observation that while probabilistic methods are often necessary for approximating shortest paths, there seems to be no justification for failing to discover the true shortest path in small-scale environments such as 11x11 mazes. This work focuses on applications where such environments suffice and suboptimal paths are unacceptable. Although our architecture supports arbitrary maze sizes for both training and testing, we concentrated on exploring the complexity introduced by varying the number of terminals rather than increasing the maze size. Scaling to larger mazes is beyond the scope of this work but represents an exciting direction for future exploration.
>
> 2. "Comparison with Modern Algorithms": We sincerely thank the reviewer for their thoughtful recommendation to include additional modern approaches, such as learning-based and reinforcement learning methods. We will carefully revise the manuscript to incorporate insights from [4] and other relevant recent methods as valuable considerations for further direction. Additionally, we wish to clarify that methods such as [1], [2], and [3] are specifically designed for input layouts in VLSI design and are not directly adapted to maze-like settings. This distinction will be explicitly highlighted in the revised version to provide a clearer understanding of the scope and focus of our work. Our primary emphasis in this paper is on achieving deterministic accuracy in small-scale environments, a feature that sets our approach apart from probabilistic methods. We will ensure this distinction is elaborated in the revised manuscript and greatly appreciate the reviewer’s guidance in helping us strengthen our work.
>
>
> 3. "TC Threshold (0.65)": The threshold value of 0.65 is a hyperparameter that was empirically chosen to achieve 100% accuracy on the test set. Although we did not perform a detailed ablation study of different thresholds, we observed that this value consistently produced the desired results in our experiments. We agree that further analysis of this parameter could provide additional insights and will include a discussion of this in future work.
>
>
> 4. "Tree Extension and Binary Matrix": Thank you for the opportunity to clarify this point. MazeNet operates as an image-processing technique rather than a traditional graph-based algorithm. As such, trees are extended irregularly during each iteration based on model predictions. Consequently, the notion of a specific number of cells added per iteration does not directly apply. This distinction highlights a fundamental departure from traditional graph-based algorithms, and we will revise the manuscript to make this explicit.
>
> 5. "Performance on Large Mazes (e.g., 256x256)": While MazeNet theoretically supports larger maze sizes, this work focuses on small, low-complexity environments where deterministic accuracy is critical. The scalability of MazeNet to larger mazes is an intriguing direction for future research. For the purposes of this work, however, we emphasize its effectiveness in small, maze scenarios. We will include a discussion of this limitation and future directions in the revised manuscript.

---

### Official Review · Reviewer_TeqD · 2024-11-06

**Soundness:** 1
**Presentation:** 2
**Contribution:** 1
**Rating:** 1
**Confidence:** 4

**Summary:**

The authors propose a neural network-based framework named Mazenet for the Obstacle Avoiding Rectilinear Steiner Minimum Tree problem, an important combinatorial problem associated with circuit routing.

Mazenet is derived from an image classification perspective. The algorithm involves mapping an input graph and set of terminals to an image. An recurrent convolutional network is then trained on synthetic data to sequentially predict elements of the steiner tree. A termination condition module is trained to detect once a candidate path is detected.

The authors demonstrate that Mazenet recovers the OARSMT faster than classical exact algorithms and highlight its ability to generalize to problem settings beyond its training set. Some ablation experiments detailing Mazenet’s  test accuracy and training time are provided. Superior runtimes are reported and perfect test accuracy.

**Strengths:**

- The authors propose a novel image-based pipeline for the OARSMT problem
- The synthetic dataset generation is interesting
- Superior runtimes are reported on a variety of synthetic benchmarks compared to classic methods

**Weaknesses:**

- weak experimental results. The authors evaluate their method on synthetic benchmarks and compare to old methods.
- some confusing results. figure 14 does not imply perfect test accuracy despite the claims made in the paper.
- the authors may consider a more rigorous evaluation with the current state of the art, FLUTE or any number of other recent methods, e.g. Chen et al., A Reinforcement Learning Agent for Obstacle-Avoiding Rectilinear Steiner Tree Construction, 2022, Kahng et al., NN-Steiner: A Mixed Neural-algorithmic Approach for the Rectilinear Steiner Minimum Tree Problem, 2023, etc.
- evaluation on real datasets is critical to understand the performance benefit of the proposed method.

**Questions:**

can the authors comment on how does the method compare to other recent works?

can the authors clarify the discrepancy between figure 14 and the perfect accuracy claims made in the main text

_our method reaches the solution in very few iterations, as seen in Figure 15. This contrasts with the competing methods, which often rely on loops that repeat for many more iterations to arrive to a solution_ - I could not understand the significance of this claim. Can the authors provide additional insight?

---

> ### Author Response · Authors · 2024-11-27
> **Responses to Reviewer TeqD Comments**
>
> 1. "Synthetic Benchmarks": Thank you for your detailed feedback regarding the evaluation on synthetic benchmarks and comparisons with recent methods. We originally defined the performance metric of accuracy as whether a method could find the shortest path for a given environment. MazeNet's motivation, which we acknowledge was not sufficiently clarified in the original paper, was rooted in addressing a gap in the literature. While probabilistic methods are often necessary for approximations of shortest paths, we believe that for a small 11x11-scale environment, discovering the true shortest path should be achievable. Thanks to your insightful comments, we have now included the wirelength ratio comparison in our analysis. However, we recognize that the wirelength ratio alone does not fully capture the operational significance of errors in determining the shortest path. To address this, we have also included an analysis of the average percentage increase in wirelength, conditioned on the occurrence of approximation errors. This addition will provide a more comprehensive evaluation of the method.
>
> 2. "Compare to old methods": We appreciate the reviewer highlighting recent literature and will cite and discuss each of these methods in the revised paper. While these contributions are important and solve related problems, they do not directly tackle our specific Obstacle-Avoiding Rectilinear Minimum Spanning Tree (OARMST) problem on planar, unweighted grid graphs with obstacles. That said, we will integrate in future work FLUTE as a benchmark due to its well-known suitability for such graphs. We also considered methods such as Chen et al.’s "A Reinforcement Learning Agent for Obstacle-Avoiding Rectilinear Steiner Tree Construction" and Kahng et al.’s "NN-Steiner: A Mixed Neural-algorithmic Approach for the Rectilinear Steiner Minimum Tree Problem." While these approaches are significant, Chen et al.’s method involves transforming layouts into weighted graphs, which diverges from our problem setting. Similarly, Kahng et al.’s method does not support obstacles, which limits its relevance to maze-like scenarios. Nonetheless, we will study these methods for inspiration in extending our work to related problems.
>
> 3. "Evaluation on real-world datasets": We recognize the importance of testing our method on real-world datasets. While our current focus has been on maze-like scenarios, we are exploring adaptations of our approach to other type of data, such as layout images, as suggested. For maze-like settings, we believe our method remains particularly suitable due to its consistent achievement of optimal solutions without errors and its demonstrated scalability.
>
> 4. "Figure 14 and accuracy discrepancy": Thank you for pointing out the potential confusion regarding Figure 14. To clarify, this figure shows the accuracy without the application of the Termination Condition (TC) module. The TC module is only applied during testing, where it ensures 100% accuracy, as presented in Tables 1 and 2. During training, the TC module is not applied, which explains why training accuracy does not reach 100%, as observed in Figure 6b. We will update the text to make this distinction clearer.
>
> 5. "Iterations": We appreciate the opportunity to clarify this aspect. The statement refers to the high-level module diagram abstraction of our approach, where the solution is achieved through a smaller number of abstract modules. We will ensure this is better explained in the revised version.

---

### Author Response · Authors · 2024-11-28

We inform the reviewers that the main issues raised have been addressed in the revised manuscript. However, due to the expanded content, the revised paper now exceeded the 10-page limit. As a result, the runtime parallelization figure, previously labeled as Figure 9, has been moved to the appendix and is now labeled as Figure 14. The figure has been properly referenced in the main text to fulfill the length requirement. We also updated the abstract to clarify our motivation for MazeNet, emphasizing that while a probabilistically correct method may be the only realistic approach for approximating shortest paths, there is no justification for not discovering the true shortest path in a small-scale environment. We thank the reviewers for their helpful comments. We will include the necessary clarifications and corrections in the revised version, which we believe will address their concerns.

---

### Meta-Review · Area_Chair_Ptpd · 2024-12-12

**Metareview:**

This paper addresses a combinatorial problem with machine learning. It received five unanimously highly critical reviews and there was a consensus on its weaknesses (Lack of novelty (overlap with another method), weak evaluation: synthetic and simple benchmarks and comparison to old baselines, confusing results, lack of positioning, unclear motivation, lacking presentation, unclear technical details).

The authors attempted to provide answers but could not solve the paper's problem.

**Additional Comments On Reviewer Discussion:**

There was nothing to discuss

---

### Decision · Program_Chairs · 2025-01-22

Reject